Correlation between long-term glycemic variability and cognitive function in middle-aged and elderly patients with type 2 diabetes mellitus: a retrospective study

Ding JingCheng 1
Shi Qian 1
Tao Qian 1
Su Hong 2
Du Yijun 1
Pan Tianrong 1
Zhong Xing zhongxing761@163.com 1
1 Department of Endocrinology, The Second Affiliated Hospital of Anhui Medical University , Hefei , Anhui Province , China
2 Department of Epidemiology and Health Statistics, Anhui Medical University , Hefei , Anhui Province , China
Liu Feng
Electronic publication date: 2023 Dec 20
Publication date: 2023
Volume: 11
Electronic Location ID: e16698
Received 2023 Aug 23; Accepted 2023 Nov 29
Copyright: ©2023 Ding et al.
Copyright year: 2023
Copyright holder: Ding et al.
License: This is an open access article distributed under the terms of the Creative Commons Attribution License, which permits unrestricted use, distribution, reproduction and adaptation in any medium and for any purpose provided that it is properly attributed. For attribution, the original author(s), title, publication source (PeerJ) and either DOI or URL of the article must be cited.
License URL: https://creativecommons.org/licenses/by/4.0/

Keywords: Type 2 Diabetes mellitus, Blood glucose variability, Cognitive function

Funding: Clinical Research Incubation Program of the Second Affiliated Hospital of Anhui Medical University 2021LCZD14 Anhui Medical University Research Fund Project Funding 2020xkj200 Anhui Medical University Endocrinology-Epidemiology Health Statistics Biochemistry Co-Construction Program 2021lcxk024 This work was supported by the Clinical Research Incubation Program of the Second Affiliated Hospital of Anhui Medical University (2021LCZD14)), the Anhui Medical University Research Fund Project Funding (2020xkj200), and the Anhui Medical University Endocrinology-Epidemiology Health Statistics- Biochemistry Co-Construction Program (2021lcxk024). The funders had no role in study design, data collection and analysis, decision to publish, or preparation of the manuscript.

==============================
Objective

To investigate the correlation between long-term glycemic variability and cognitive function in middle-aged and elderly patients with type 2 diabetes mellitus (T2DM).

Methods

This retrospective analysis includes 222 patients hospitalized at Second Affiliated Hospital of Anhui Medical University from June 2021 to June 2023. Cognitive function was assessed using the Montreal Cognitive Assessment (MoCA) and Mini-Mental State Examination (MMSE). All patients were categorized into the MCI group and the non-MCI group based on their MoCA score. Long-term blood glucose fluctuations were measured using glycated hemoglobin A1c standard deviation (HbA1c-SD) and fasting plasma glucose standard deviation (FPG-SD). The study compared general clinical data, blood biochemical indicators, and glycemic variability indicators between the two groups. The differences between the groups were compared using t-test, Chi-Square Test, or Mann–Whitney U test. Kendall’s correlation analysis, multivariate logistic regression analysis and ROC curve correlation analysis were further used to analyze the correlation and diagnostic power.

Results

The differences in age, MoCA scores, MMSE scores, HOMA-β, HbA1c-M, HbA1c-SD, FPG-M, FPG-SD, eGFR, Smoking, GLP-1RA and SGLT-2i usage were statistically significant between the two groups (P < 0.05). Kendall’s correlation analysis showed that age, HbA1c-M, HbA1c-SD, FPG-M, and FPG-SD was negatively correlated with MoCA scores; meanwhile, the HOMA-β, and eGFR was positively correlated with MoCA scores. Multiple logistic regression analysis revealed that HbA1c-SD, FPG-SD and Smoking were risk factors for cognitive dysfunction, while eGFR, GLP-1RA and SGLT-2i usage was a protective effect. The area under the curve (AUC) values for predicting MCI prevalence were 0.830 (95% CI [0.774–0.877], P < 0.001) for HbA1c-SD, 0.791 (95% CI [0.655–0.808], P < 0.001) for FPG-SD, and 0.698 (95% CI [0.633–0.757], P < 0.001) for eGFR. The optimal diagnostic values were 0.91, 1.32, and 74.81 ml/min/1.73 m2 for HbA1c-SD, FPG-SD, and eGFR, respectively.

Conclusions

Cognitive function in middle-aged and elderly T2DM patients is influenced by long-term blood glucose variability, with poorer cognitive function observed in individuals with higher blood glucose variability. The impact of HbA1c-SD on MCI exhibited a greater magnitude compared to that of PFG-SD and smoking. Additionally, renal function, GLP-1RA and SGLT-2i usage exert positive effects on cognitive function.

Introduction

Diabetes mellitus is a metabolic disease characterized by high blood sugar levels. In China (Li et al., 2020), the prevalence of Type 2 Diabetes Mellitus (T2DM) among adults is as high as 12.8%. Diabetes can cause severe damage to various organs, such as the heart, brain, and kidneys, and can be life-threatening. Diabetes cognitive dysfunction is one of the complications of hyperglycemia involving the central nervous system, and T2DM is also thought to be an independent risk factor for cognitive dysfunction (Xue et al., 2019). There are two forms of diabetic hyperglycemia: persistent and fluctuating. The International Diabetes Federation guideline states that acute fluctuating hyperglycemia is more harmful than chronic persistent hyperglycemia (International Diabetes Federation Guideline Development Group, 2014). The greater the fluctuations in blood glucose levels, the higher the risk of chronic complications. Patients with T2DM who achieve standard glycemic control show better cognitive performance on tests like cognitive assessment, Trail Making Test-B, and verbal fluency test compared to those with greater glycemic fluctuations (Xia et al., 2020).

Blood glucose variability (Škrha et al., 2016) refers to the unstable state of blood sugar levels between maximum and minimum values. The effects of glucose variability on cognitive function can be monitored through short-term continuous glucose monitoring systems. However, the effects of long-term glucose fluctuations on cognitive function in middle-aged and older Chinese patients are rarely reported. This study aims to investigate the effects of long-term glycemic variability on cognitive function and other related factors in middle-aged and older Chinese patients with T2DM through a retrospective survey study. The findings will provide insights for future clinical treatment.

Materials and Methods

Portions of this text were previously published as part of a preprint (https://assets.researchsquare.com/files/rs-2504178/v1_covered.pdf?c=1681796690).

Research population

This retrospective study included 222 middle-aged and elderly patients with T2DM who were regularly visited at the Second Affiliated Hospital of Anhui Medical University between June 2021 and June 2023. Inclusion criteria were the following: (1) Type 2 diabetes mellitus diagnosed based on the statement of the American Diabetes Association (ADA) in 2011 (American Diabetes Association, 2011); (2) they were between 50 and 85 years old; (3) they were no clear cerebral infarction and extensive cerebral white matter demyelination on head magnetic resonance examination. Exclusion criteria were the following: (1) previous clear diagnosis of dementia, stroke, Parkinson’s, and other diseases affecting cognitive function; (2) acute and chronic severe diabetic complications (DKA, HHS, diabetic foot); (3) major medical illness, such as severe heart disease, and damaged liver or kidney function, tumors and chronic infections, blood and immune system diseases, psychiatric Patients, especially those who need to take hormones. (4) Severe hearing, visual impairment or physical disability.

The protocol and informed consent document were approved by the research ethics committees of the Second Affiliated Hospital of Anhui Medical University (Approved No. of ethic committee: YX2022-043).

Demographic and clinical information

We collected demographic and clinical data from all patients, including age, gender, height, weight, SBP, DBP, duration of diabetes, educational background, glucose-lowering drugs, smoking history, family history of diabetes, coronary vascular disease, etc. Blood tests were carried out after an overnight fasting for the current fasting plasma glucose (FPG), glycosylated hemoglobin A1c (HbA1c), fasting C-peptide (FC-P), alanine aminotransferase (ALT), aspartate aminotransferase (AST), total cholesterol (TC), creatinine, 24-hour urinary protein excretion, urine albumin-to-creatinine ratio (UACR), thyroid-stimulating hormone (TSH) and other relevant parameters. Patients’ BMI were calculated. All participants visited our endocrinology outpatient clinic at 3-month intervals, and the fasting plasma glucose (FPG) and glycated hemoglobin A1c (HbA1c) values were obtained at the visits. Based on the recorded Glycated hemoglobin A1c (HbA1c), fasting plasma glucose (FPG) at each visit, mean glycated hemoglobin A1c (HbA1c-M) and standard deviation (HbA1c-SD), mean fasting plasma glucose (FPG-M) and standard deviation (FPG-SD) were calculated, respectively. And the standard deviation (SD) of HbA1c and FPG were used as indicators of glucose variability in the present study.

Homeostasis model assessment (HOMA- β) and Homeostasis model assessment insulin resistance (HOMA-IR) were calculated by HOMA2 calculator (HOMA2 v2.2.3; https://www.dtu.ox.ac.uk/homacalculator/). Calculation of patient surface area (BSA) based on the DuBois formula (Du Bois & Du Bois, 1989): 0.007184 × weight (kg)0.42 5 × height (cm)0.725. Calculation of creatinine clearance rate (Ccr) using Cockcroft-Gauh (CG) formula (Cockcroft & Gault, 1976): eCcr for men = (140-age) × weight/ (72 × Scr); eCcr for women = 0.85 × (140-age) × weight/(72 × Scr); according to gender, GFR calculated by the Cockcroft-Gauh (CG) formula was normalized by BSA: GFR[ml/min/1.73 m2] = 0. 84 × eCcr × (1.73/BSA). The unit of blood creatinine in the above formula is mg/dl, and 1 µmol/l = 0.0113 mg/dl.

Cognitive function assessment

Evaluators are rigorously trained and a uniform survey scale is used. All enrolled patients were evaluated on the Montreal Cognitive Assessment Scale (MoCA) and the Mini-Mental State Examination (MMSE). Both cognitive tests covers distinct cognitive skills, and scores range from 0 to 30. This study refer to the findings of Professor Jia’s research conducted on middle-aged and elderly individuals within the community during the Eleventh Five-Year Plan, taking into full consideration the influence of educational attainment on assessment outcomes, and establishing standardized criteria for cognitive decline. MoCA criteria (Lu et al., 2011) for cognitive impairment: Illiterate education ≤ 13 points, elementary school education ≤ 19 points, junior high school education and above ≤ 24 points. MMSE criteria (Li, Jia & Yang, 2016) for cognitive impairment: Illiterate education ≤ 19 points, elementary school education ≤ 22 points, junior high school education and above ≤ 26 points. All patients were categorized into the MCI group and the non-MCI group based on their MoCA score.

Statistical methods

All statistical analyzes using SPSS version 26.0 software (SPSS Inc., Chicago, IL, USA). Comparisons between groups were made using t-test for normally distributed variables, the Mann–Whitney U test for asymmetrically distributed variables, and chi-square test for categorical variables. For all participants, Kendall’s correlation analysis was used to assess the relationship between MoCA scores and clinical indicators. Variables with statistical differences in the univariate analysis were then entered into a multivariable logistic regression analysis model to determine their net effects on cognitive function. Odds ratios and their 95% confidence intervals (CIs) were used to assess the independent contribution of prognostic factors. Using (1-specificity) as transverse coordinates and sensitivity as longitudinal coordinates, the subjects’ working characteristic curves (ROCs) were mapped and the area under the ROC curve was calculated (AUC) using MedCalc Software version 20.100 (https://www.medcalc.org/). A two-sided P-value <0.05 was considered to indicate statistical significance.

Results

Participant characteristics

The participants’ clinical characteristics are summarized in Table 1. Among the 222 middle-aged and elderly patients with T2DM, the MCI group consisted of 77 males and 35 females, with an average visit of 16.1 months. The control group included 76 males and 34 females, with an average visit of 16.4 months. Compared with the control group, MOCA scores, MMSE scores, eGFR, HOMA- β, GLP-1RA and SGLT-2i usage in the MCI group were significantly decreased (P < 0.05); and age, HbA1c-M, HbA1c-SD, FPG-M, FPG-SD and smoking rate were significantly increased (P < 0.05). However, there were no statistically significant differences in gender, education levels, duration of diabetes, family history of diabetes, coronary vascular disease, BMI, BSA, SBP, DBP, TC, ALT, AST, TSH, 24 h urinary protein, UACR, other glucose-lowering drugs, etc (P > 0.05).

Table 1 Characteristics of the study cohort.

We conducted a comprehensive analysis of demographic and clinical disparities between the MCI and control groups.

Projects	Non-MCI group
(n = 110)	MCI group
(n = 112)	P value	
Age (years)	60.73(56.00–65.00)	63.07(57.00–69.50)	0.029	
Gender (Male/Female)	76/34	77/35	0.420	
Education n(%)				
Illiteracy	5(4.5%)	9(8.0%)	0.956	
Primary School	25(22.8%)	20(17.9%)		
Junior high school and above	80(72.7%)	83(74.1%)	
MoCA Scores (minutes)	25.12(24.75–27.00)	20.88(19.00–23.00)	<0.001	
MMSE Scores (minutes)	26.56(25.00–28.00)	23.61(22.00–25.00)	<0.001	
Diabetic duration (years)	13.02(8.00–17.25)	14.37(9.25–19.75)	0.170	
SBP (mmHg)	125.56 ± 14.05	128.57 ± 14.38	0.116	
DBP (mmHg)	73.97(68.00–79.00)	74.96(67.00–80.75)	0.503	
BMI (kg/m2)	25.47 ± 2.97	24.98 ± 2.83	0.209	
BSA (m2)	1.79 ± 0.16	1.79 ± 0.18	0.995	
FC-P (ng/ml)	2.26(1.59–2.74)	2.12(1.42–2.61)	0.219	
HOMA- β (%)	81.48(53.55–98.23)	65.37(39.25–85.48)	<0.001	
HOMA-IR (%)	1.84(1.27–2.20)	1.79(1.23–2.15)	0.540	
HbA1c-Mean (%)	7.46(6.59–8.03)	8.31(7.49–8.88)	<0.001	
HbA1c-SD	0.60(0.43–0.75)	1.12(0.77–1.41)	<0.001	
FPG-Mean (mmol/L)	7.02(6.30–7.57)	7.99(6.96–8.94)	<0.001	
FPG-SD	0.92(0.62–1.21)	1.56(1.07–2.10)	<0.001	
TC (mmol/L)	4.32(3.58–4.86)	4.35(3.70–4.94)	0.840	
ALT (U/L)	19.65(14.00–24.25)	18.37(14.00–22.00)	0.186	
AST (U/L)	20.56(17.00–23.00)	19.97(17.00–23.00)	0.615	
TSH (mIU/L)	2.16(1.28–2.77)	2.24(1.29–2.91)	0.704	
eGFR (ml/min/1.73m2)	87.78 ± 18.96	74.28 ± 17.85	<0.001	
24 h urinary protein (mg/L)	81.02(51.89–93.85)	89.36(62.68–113.50)	0.065	
UACR (mg/g)	24.89(13.38–28.74)	23.45(13.95–26.83)	0.358	
Family history of diabetes n(%)	40(36.4%)	34(30.4%)	0.343	
Coronary heart disease n(%)	22(20.0%)	26(23.2%)	0.561	
Smoking n(%)	30(27.3%)	53(47.3%)	0.002	
GLP-1RA n(%)	43(39.0%)	29(25.8%)	0.036	
Insulins n(%)	36(32.7%)	43(38.3%)	0.378	
Insulin secretagogues n(%)	17(15.4%)	24(21.4%)	0.251	
Thiazolidinediones n(%)	13(11.8%)	14(12.5%)	0.877	
SGLT-2 Inhibitors n(%)	47(42.7%)	31(27.6%)	0.019	
DPP-4 inhibitors n(%)	29(26.3%)	35(31.2%)	0.422	
a-glucosidase inhibitors n(%)	26(23.6%)	25(22.3%)	0.816	
Biguanides n(%)	39(35.4%)	45(40.1%)	0.686	
Notes.

MoCA the Montreal Cognitive Assessment

MMSE the Mini-Mental State Examination

SBP systolic blood pressure

DBP diastolic blood pressure

BMI body mass index

BSA body surface area

FC-P fasting C-peptide

HOMA- β islet function index

HOMA-IR insulin resistance index

HbA1c-Mean the mean glycated hemoglobin A1c

HbA1c-SD the standard deviation of glycated hemoglobin A1c

FPG-Mean mean fasting plasma glucose

FPG-SD the standard deviation of fasting plasma glucose

TC total cholesterol

ALT alanine aminotransferase

AST aspartate aminotransferase

eGFR expected glomerular filtration rate

UACR urinary micro-albumin/creatinine ratio

TSH thyroid stimulating hormone

GLP-1RA Glucagon-like peptide-1 receptor agonists

SGLT-2i sodium-glucose co-transporter type-2 inhibitors

DPP-4 inhibitors dipeptidyl peptidase 4 inhibitors

Values are expressed as mean ±standard deviation (SD) or median (range 25th–75th percentile).

Associations of cognitive dysfunction with clinical indicators

Kendall’s correlation analysis showed a negative correlation between age (b = −0.162, p = 0.001), HbA1c-M (b = −0.212, p < 0.001), HbA1c-SD (b = −0.323, p < 0.001), FPG-M (b = −0.215, p < 0.001), and FPG-SD (b = −0.225, p < 0.001) and MoCA scores. Meanwhile the HOMA- β (b = 0.144, p = 0.002) and eGFR (b = 0.174, p < 0.001) was positively correlated with MoCA scores (Table 2).

Table 2 Kendall’s correlation analyses of cognitive function and clinical indicators.

The correlation between MoCA scores and clinical indicators was examined in our analysis.

Group	Age	HOMA-β	HbA1c-Mean	HbA1c-SD	FPG-Mean	FPG-SD	eGFR	
MoCA Scores								
b	−0.162	0.144	−0.212	−0.323	−0.215	−0.225	0.174	
P value	0.001	0.002	<0.001	<0.001	<0.001	<0.001	<0.001	
Notes.

HOMA- β islet function index

HbA1c-Mean mean glycated hemoglobin A1c

HbA1c-SD standard deviation of glycated hemoglobin A1c

FPG-Mean mean fasting plasma glucose

FPG-SD standard deviation of fasting plasma glucose

eGFR expected glomerular filtration rate.

Univariate logistic regression analysis showed a statistically significant difference in age, HOMA- β, HbA1c-M, HbA1c-SD, FPG-M, FPG-SD, eGFR, smoking, GLP-1RA usage between the MCI and Non-MCI group. After adjusting for age, HOMA- β and other factors in multivariate model, HbA1c-SD (OR = 77.657, 95% CI [15.064–400.334], P < 0.001), FPG-SD (OR = 9.945, 95% CI [3.602–27.458], P < 0.001), and smoking (OR = 2.874, 95% CI [1.185–6.969], P < 0.05) remained independently associated with MCI. Similarly, eGFR level (OR = 0.943, 95% CI [0.916–0.970], P < 0.001), GLP-1RA usage (OR = 0.226, 95% CI [0.082–0.623], P < 0.05), and SGLT-2i usage (OR = 0.337, 95% CI [0.135–0.843], P < 0.05) was also independently associated with MCI (Table 3).

Table 3 Univariate and multi-factor logistic regression analysis of factors influencing cognitive dysfunction in T2DM patients.

We conducted univariate and multivariate logistic regression analyses to investigate the factors influencing cognitive dysfunction in patients with type 2 diabetes mellitus.

	Univariate analysis	Multi-Factor Analysis	
	OR	95% CI	P	OR	95% CI	P value	
Age	1.052	1.011–1.094	0.012	0.953	0.883–1.030	0.223	
HOMA- β	0.988	0.980–0.995	0.002	0.990	0.975–1.005	0.183	
HbA1c-Mean	1.839	1.422–2.378	<0.001	0.955	0.609–1.499	0.843	
HbA1c-SD	55.350	18.561–165.054	<0.001	77.657	15.064–400.334	<0.001	
FPG-Mean	1.700	1.362–2.123	<0.001	1.049	0.698–1.577	0.818	
FPG-SD	8.798	4.600–16.828	<0.001	9.945	3.602–27.458	<0.001	
eGFR	0.961	0.946–0.976	<0.001	0.943	0.916–0.970	<0.001	
Smoking	2.395	1.368–4.194	0.002	2.874	1.185–6.969	0.020	
GLP-1RA	0.544	0.308–0.963	0.037	0.226	0.082–0.623	0.004	
SGLT-2i	0.513	0.293–0.898	0.020	0.337	0.135–0.843	0.020	
Notes.

HOMA- β islet function index

HbA1c-Mean mean glycated hemoglobin A1c

HbA1c-SD standard deviation of glycated hemoglobin A1c

FPG-Mean mean fasting plasma glucose

FPG-SD standard deviation of fasting plasma glucose

eGFR expected glomerular filtration rate

GLP-1RA Glucagon-like peptide-1 receptor agonists

SGLT-2i sodium-glucose co-transporter type-2 inhibitors

ROC curve analysis of the predictive value of HbA1c -SD, FPG-SD, and eGFR for MCI prevalence

The area under the curve (AUC) of HbA1c-SD for predicting MCI prevalence was 0.830 (95% CI [0.774–0.877], P < 0.001), with a sensitivity of 68.70%, a specificity of 90.00%, and an optimal diagnostic value 0.91 (Fig. 1A). The area under the curve (AUC) of FPG-SD for predicting MCI prevalence was 0.791 (95% CI [0.655–0.808], P < 0.001), with a sensitivity of 60.70%, a specificity of 82.70%, and an best diagnostic value 1.32 (Fig. 1A).The area under the curve (AUC) of eGFR for prediction of MCI prevalence was 0.698 (95% CI [0.633–0.757], P < 0.001), with a sensitivity of 58.90%, a specificity of 74.50% and an optimal diagnostic value 74.81 ml/min/1.73m2 (Fig. 1B).

Figure 1 ROC curves for HbA1c-SD, FPG-SD and eGFR.

(A) The predictive value of HbA1c-SD and FPG-SD in MCI. (B) The predictive value of eGFR in MCI.

Discussion

This study aimed to explore the long-term glycemic variability and other influencing factors with cognitive function in middle-aged and elderly individuals diagnosed with type 2 diabetes. Glycemic variability was assessed by HbA1c-SD and PFG-SD. The multivariable logistic regression revealed a positive correlation between higher levels of HbA1c-SD and PFG-SD, and an increased risk of cognitive impairment in those patients. Furthermore, smoking habits emerge as an independent risk factor for mild cognitive impairment (MCI) in middle-aged and elderly patients with T2DM, while the eGFR level, GLP-1RA and SGLT-2i usage may exert a protective effect.

In this study, the Montreal Cognitive Assessment Scale (MoCA) and Mini-Mental State Examination (MMSE) were employed to evaluate the cognitive function of middle-aged and elderly individuals diagnosed with T2DM. The Mini-Mental State Examination Scale (MMSE), developed in 1975, has undergone various translations and is extensively employed in both clinical practice and epidemiological research. In 1998, Katzman’s team conducted an epidemiological survey in Shanghai, China, fully considering the cultural differences between China and foreign countries, adapted the original MMSE scale and formed a Chinese version. The Montreal Cognitive Assessment Scale (MoCA), developed by Dr. Nasreddine in 2005, encompasses a comprehensive evaluation of cognitive abilities with a maximum score of 30. This assessment primarily encompasses eight domains including executive function, attention and memory, calculation and orientation, language proficiency, visuospatial perception, and abstract reasoning. Jia et al. (2021) discovered that MMSE and MoCA had good correlation and consistency in the cognitive screening of middle-aged and elderly individuals. However, the ceiling effect of the Montreal Cognitive Assessment (MoCA) on mild cognitive impairment (MCI) is minimal, and it demonstrates good test-re well as high sensitivity to subtle brain abnormalities. Consequently, the results obtained from MoCA evaluation serve as the basis for grouping in this study.

Healthy individuals exhibit normal islet secretion function and maintain a stable blood glucose fluctuation curve throughout the day. Conversely, in patients with type 2 diabetes mellitus resulting from either islet failure or insulin resistance, alterations in the internal glycemic environment not only lead to a significant increase in average blood glucose levels but also amplify the amplitude of blood glucose fluctuations. Notably, even among diabetics with similar HbA1c levels, there can be substantial variations in the magnitude of blood sugar fluctuations. Continuous glucose monitoring systems (CGMS), as an emerging method for blood glucose monitoring, offers uninterrupted, comprehensive, and reliable information on daily glycemic levels. Consequently, it has gained widespread adoption in the clinical diagnosis and treatment of individuals with diabetes mellitus.

Cui et al. (2014) employed continuous glucose monitoring systems (CGMS) to assess short-term blood glucose variability, revealing a positive correlation between increased blood glucose variability and pronounced brain atrophy as well as cognitive function deterioration in patients with T2DM. Rizzo et al. (2010) reported a positive correlation between daily acute blood glucose fluctuations and an elevated risk of dementia too. The CGMS solely captures short-term fluctuations in blood glucose levels, whereas long-term glycemic variability (Škrha et al., 2016) refers to the trend of HbA1c or FPG changes between consecutive clinic visits, encompassing a longer time frame than intra-day, daytime, and weekly blood glucose measurements. Consequently, it provides a more comprehensive reflection of glycemic variability. The HbA1c-SD is considered a valuable indicator for assessing the variability of HbA1c (Cavalot, 2013), while PFG-SD also exhibits a close association with microvascular complications (Zhou et al., 2020).

In this study, it was found that the levels of HbA1c-SD and PFG-SD in the MCI group were significantly higher than those in the control group. After adjusting for age, HOMA- β and other factors in multivariate model, HbA1c-SD and PFG-SD still independently served as risk factors for cognitive function. (OR = 77.657, 95% CI [15.064–400.334], P < 0.001), OR = 9.945, 95% CI [3.602–27.458], P < 0.001, respectively). Kim et al. (2015) observed a significant association between higher levels of HbA1c-SD and HbA1c-CV and impaired cognitive function test performance in elderly diabetic patients. Zheng et al. (2021) investigated the blood glucose variability in middle-aged and elderly individuals with diabetes in the UK from 1987 to 2018, revealing a significant association between substantial fluctuations in HbA1c levels and an increased risk of dementia among middle-aged and elderly patients with type 2 diabetes. In this study, the cut-off values for HbA1c-SD and FPG-SD were determined to be 0.91 and 1.32, respectively. When the level of HbA1c-SD and FPG-SD exceeds these values, it may play a more significant role in the occurrence of MCI in T2DM patients.

Blood glucose fluctuations, known as glycemic variability, have been found to increase the risk of complications in those with diabetes. These studies have demonstrated that glycemic variability is strongly associated with adverse events such as cardiovascular disease (Shen et al., 2021), peripheral neuropathy (Yang et al., 2021) and high mortality (Tseng et al., 2020) in T2DM patients. The adverse effects of glycemic fluctuations on cognitive function may involve oxidative stress and inflammatory damage. Hsieh et al. (2019) demonstrated that abnormal blood glucose fluctuations can cause oxidative stress in murine microglial cells, leading to ongoing neurodegenerative damage. Another study (Wang et al., 2021) found that acute glycemic fluctuations can increase the expression of inflammatory factors like TNF-a and IL-1a, which contribute to neuronal apoptosis in the hippocampus. This impairs the integrity of the blood–brain barrier and worsens central neurodegeneration. Additionally, abnormal glycemic fluctuations can also impact polyneuropathy, renal function, islet function, and insulin resistance (Yang et al., 2021; Weiner et al., 2017; Dimova et al., 2019), all of which can affect cognitive function in diabetic patients.

The patients enrolled in this study exhibited glomerular filtration rates at stage 3 or higher (eGFR >30 ml/min/1.73m2). Data on 24-hour urinary protein as well as the urine albumin-to-creatinine ratio were also collected. The multivariate logistic regression analysis revealed that maintaining a high estimated glomerular filtration rate (eGFR) could potentially serve as a protective factor for MCI in middle-aged and elderly patients with T2DM (OR = 0.943, 95% CI [0.916–0.970], P < 0.001). The glomerular filtration rate (GFR), which measures kidney function, is an important indicator. Bai et al. (2017) observed an increased risk of cognitive dysfunction in Chinese elderly individuals with mild to moderate eGFR decline compared to those with normal renal function. Dyer et al. (2022) found a robust correlation between impaired kidney function and neuropsychological disorders in individuals with an eGFR less than 45 ml/min/1.73 m2. Berger et al. (2016) also found that cognitive changes occur early in CKD, with notable impact on orientation, attention, and language proficiency. The findings of previous studies (Low et al., 2017) have revealed that HbA1c variability is strongly and independently associated with eGFR decline in T2DM patients independent of mean HbA1c. According to several clinical trials (UKPDS, ACCORD, and VADT), Zhou et al. (2020) found that there is a link between fasting plasma glucose (FPG) variability and an increased risk of moderate-to-severe diabetic nephropathy. Chiu et al. (2020) found that higher HbA1C variability is more likely to progress to macroalbuminuria in those patients who are already in a microalbuminuria state. Therefore, prolonged blood glucose variability results in the deterioration of renal function and cognitive impairment, while also creating a vicious cycle between renal and cognitive function.

In order to mitigate the impact of glucose-lowering medications on the outcomes, hypoglycemic medications for enrolled patients were categorized into eight groups in this study: GLP-1 receptor agonists, insulins, insulin secretagogues, SGLT-2 inhibitors, DPP-4 inhibitors, thiazolidinediones, a-glucosidase inhibitors, and biguanides. The result showed a significantly higher proportion of GLP-1 receptor agonists and SGLT-2 Inhibitors in the cognitively normal group compared to the MCI group, and multivariable logistic regression analysis suggested that Use of GLP1 analogs and SGLT2 inhibitors were associated with lower odds of MCI after multible adjustments (OR = 0.226, 95% CI [0.082–0.623], P < 0.05). The glucagon-like peptide-1 (GLP-1) receptor agonist, as an incretin, exerts its glucose-lowering effects through glucose-dependent stimulation of insulin secretion and inhibition of glucagon release. Additionally, it exhibits favorable impacts body weight loss, and maintenance of blood pressure stability. The REWIND study (Cukierman-Yaffe et al., 2020) revealed that long-term treatment with GLP-1RA dulaglutide might reduce cognitive impairment in people with type 2 diabetes mellitus. Gejl et al. (2016) found that GLP-1RA liraglutide exhibited significant enhancements in cognitive function among patients with MCI and AD, as compared to the placebo group. Several basic researches (Talbot & Wang, 2014; Batista et al., 2019; Tai et al., 2018) found that GLP-1 receptor agonists possess the ability to effectively retard neurodegeneration in Alzheimer’s disease, potentially through their mechanism of attenuating neuroinflammation and reducing amyloid deposition.

In addition, SGLT-2 inhibitors could potentially serve as a protective factor against diabetes-associated cognitive impairment in our model (OR = 0.337, 95% CI [0.135–0.843], P < 0.05). Pawlos et al. (2021) found that SGLT2 inhibitors are fat-soluble and cross the blood–brain barrier to reach various parts of the brain. Animal studies (Hierro-Bujalance et al., 2020) showed empagliflozin reduces vascular damage and cognitive impairment in APP/PS1xdb/db mice. Wium-Andersen et al. (2019) found a significant reduction in the risk of dementia among T2DM patients treated with SGLT2 inhibitors compared to the control group. However, a prospective study (Cheng et al., 2022) demonstrated that demonstrated that liraglutide enhanced impaired brain activation and restored impaired cognitive domains in patients with type 2 diabetes, whereas dapagliflozin did not. The disparities observed in the aforementioned study findings may be attributed to variations in the specific SGLT-2i agents, age distribution and duration of diabetes among the enrolled patients.

Restifo et al. (2022) found in their study that smoking can be considered as a risk factor for cognitive dysfunction in middle-aged and elderly people. Lewis et al. (2021) demonstrated that smoking had a more detrimental impact on the cognitive abilities of women compared to men. Zhang et al. (2016) research revealed that smoking is associated with cognitive decline and smoking severity is positively associated with BDNF levels in Chinese Han population. Nicotine is one of the primary substances that elicit dependence in cigarettes. Alhowail (2021) showed that nicotine affects cognitive development by altering the expression and function of nAChR subunits, altering ERK1/2 activity, and increasing GluR2 surface expression, as well as decreasing neurogenesis, synaptic plasticity in the hippocampus and cortex. Consistent with the above findings, this study found smoking may serve as a independent risk factor for cognitive function in middle-aged and elderly patients with type 2 diabetes mellitus. (OR = 2.874, 95% CI [1.185–6.969], P < 0.05).

However, it should be noted that our study has a few limitations. Firstly, it was a single-center retrospective design with a small sample size, which limited the statistical power of our analyses. Additionally, the age range of our participants was limited to 50–85 years, so the relationship between other age groups and blood glucose fluctuations remains unclear. We plan to address these limitations in future research.

Conclusions

Our findings suggest that long-term glucose variability, measured by HbA1c-SD and FPG-SD, is strongly linked to cognitive function in middle-aged and older individuals with T2DM. We observed that greater fluctuation in blood glucose is associated with more severe cognitive decline. Furthermore, it is imperative to consider patients’ smoking status, renal function, and utilization of hypoglycemic medications. From a clinical perspective, it is crucial to take a personalized approach and consider glycemic variability as a target for treatment when managing T2DM patients. This approach can help reduce the risk of cognitive impairment and other associated complications.

Supplemental Information

Supplemental Information 1 Raw data

Click here for additional data file.

We would like to express our heartfelt gratitude to all the staff of the Department of Endocrinology, the Second Affiliated Hospital of Anhui Medical University and Department of Epidemiology and Health Statistics, Anhui Medical University for their selfless help and valuable assistance.

Abbreviations

MoCA The Montreal Cognitive Assessment

MMSE The Mini-Mental State Examination

SBP Systolic Blood Pressure

DBP Diastolic Blood Pressure

BMI body mass index

BSA body surface area

FC-P fasting C-peptide

HOMA- β islet function index

HOMA-IR insulin resistance index

HbA1c-Mean the mean glycated hemoglobin A1c

HbA1c-SD the standard deviation of glycated hemoglobin A1c

FPG-Mean mean fasting plasma glucose

FPG-SD the standard deviation of fasting plasma glucose

TC total cholesterol

ALT alanine aminotransferase

AST aspartate aminotransferase

eGFR expected glomerular filtration rate

UACR urinary micro-albumin/creatinine ratio

TSH thyroid stimulating hormone

GLP-1RA Glucagon-like peptide-1 receptor agonists

SGLT-2i sodium-glucose co-transporter type-2 inhibitors

DPP-4 inhibitors dipeptidyl peptidase 4 inhibitors

1 mmHg = 0.133 kPa 1 mmHg = 0.133 kPa.

Additional Information and Declarations

Competing Interests

Author Contributions

Ethics

Data Availability

The authors report no conflicts of interest in this work.

JingCheng Ding conceived and designed the experiments, performed the experiments, analyzed the data, prepared figures and/or tables, authored or reviewed drafts of the article, and approved the final draft.

Qian Shi performed the experiments, prepared figures and/or tables, and approved the final draft.

Qian Tao performed the experiments, prepared figures and/or tables, and approved the final draft.

Hong Su conceived and designed the experiments, prepared figures and/or tables, and approved the final draft.

Yijun Du performed the experiments, prepared figures and/or tables, authored or reviewed drafts of the article, and approved the final draft.

Tianrong Pan analyzed the data, authored or reviewed drafts of the article, and approved the final draft.

Xing Zhong conceived and designed the experiments, performed the experiments, analyzed the data, prepared figures and/or tables, authored or reviewed drafts of the article, and approved the final draft.

The following information was supplied relating to ethical approvals (i.e., approving body and any reference numbers):

The protocol and informed consent document were approved by the research ethics committees of the Second Affiliated Hospital of Anhui Medical University(Approved No. of ethic committee: YX2022-043).

The following information was supplied regarding data availability:

The raw data is available in the Supplementary File.

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
