# Peer review of "Correlation between long-term glycemic variability and cognitive function in middle-aged and elderly patients with type 2 diabetes mellitus: a retrospective study"

_PeerJ, doi:10.7717/peerj.16698_

## Round 0.1 · original submission · Major Revisions

All the reviewers raised major concerns. The authors should carefully address them.

Reviewer 1 ·

Basic reporting

I recommend the authors to provide figures with higher resolution, ensuring that all labels, lines, and data points are clearly visible.

Experimental design

1. While glycemic variability in T2DM is an important clinical issue, much of the data presented seems confirmatory of previous findings. To make a more novel contribution, the authors could consider additional analyses, subgroup evaluations, or longitudinal follow-up to provide new information related to mechanisms, moderators, or causal relationships.
2.The retrospective design means causality cannot be inferred. Glycemic variability may potentially cause cognitive decline, but a prospective study is needed to demonstrate this. Conclusions go beyond what can be inferred from the data - causal statements about glycemic variability's effect on cognition are too strong.
3. Does not control for potential confounders like medications, cardiovascular disease, smoking that could impact cognition. Confounders should be acknowledged as limitation and controlled for in analysis.
4. The cognitive assessment relies only on the MoCA. Additional cognitive tests assessing different domains would strengthen the results.
5. Small events rates (only 110 with MCI) limit robustness of regression models

Validity of the findings

No comment.

Additional comments

Overall, this study provides useful preliminary evidence but is limited by its retrospective design, narrow sample, and lack of causal inferences. Expanding the limitations, controls, details, and tempering the conclusions would improve the paper.

Annotated reviews are not available for download in order to protect the identity of reviewers who chose to remain anonymous.

Reviewer 2 ·

Basic reporting

This study examined the relationship between long-term blood glucose variability and cognitive function in middle-aged and elderly patients with type 2 diabetes, employing various statistical methods for data analysis. However, some issues should be considered to improve the paper.

Experimental design

no comment

Validity of the findings

no comment

Additional comments

1. The diagnostic criteria for diabetes patients used in the study are based on the 1999 WHO standards, which are relatively outdated and do not align with the current realism of the research.
2. Blood glucose levels in diabetes patients are influenced by various factors, including the progression of the disease and changes in treatment plans. Appropriate statistical methods need to be employed to control for these potential confounding factors.
3. Missing data encountered in the study require reasonable handling and analysis to ensure the accuracy and reliability of research results.
4. To enhance the readability and comprehensibility of the research report, it is advisable to include relevant annotations and explanations below the tables and images. Particularly, for English abbreviations and acronyms, clear explanations should be provided to ensure that readers understand their meanings. This helps to avoid potential confusion and misunderstandings.
5. How was the MCI group distinguished from the non-MCI group in this study, and please provide a detailed description in the methods section.
6. In the discussion, please provide more information on the biological mechanisms underlying the association between cognitive function and clinical indicators or relevant prior research to support this finding.
7. In the multivariate model, HbA1c-SD, FPG-SD, and eGFR were identified as independent factors related to MCI. Please elaborate on the biological or clinical explanations for how these factors are associated with the occurrence of MCI.
8. The ROC curve analysis shows the performance of HbA1c-SD, FPG-SD, and eGFR in predicting MCI incidence. How to interpret the ROC curve's area under the curve (AUC) for these indicators, and what are their prospects for practical clinical applications?

---

## Round 0.2 · accepted · Accept

Reviewer 1 ·

Basic reporting

no comment.

Experimental design

no comment.

Validity of the findings

no comment.

Reviewer 2 ·

Basic reporting

no comment

Experimental design

no comment

Validity of the findings

no comment

Additional comments

no comment